# The medial axis of closed bounded sets is Lipschitz stable with respect to the Hausdorff distance under ambient diffeomorphisms

## Abstract

We prove that the medial axis of closed sets is Hausdorff stable in the following sense: Let $\mathcal{S} \subseteq \mathbb{R}^d$ be a fixed closed set that contains a bounding sphere. Consider the space of $C^{1,1}$ diffeomorphisms of $\mathbb{R}^d$ to itself, which keep the bounding sphere invariant. The map from this space of diffeomorphisms (endowed with a Banach norm) to the space of closed subsets of $\mathbb{R}^d$ (endowed with the Hausdorff distance), mapping a diffeomorphism $F$ to the closure of the medial axis of $F(\mathcal{S})$, is Lipschitz.

This extends a previous stability result of Chazal and Soufflet on the stability of the medial axis of $C^2$ manifolds under $C^2$ ambient diffeomorphisms.

## 1 Introduction

In [19], Federer introduced the *reach* of a (closed) set $\mathcal{S} \subset \mathbb{R}^d$ as the infimum over all points in $\mathcal{S}$ of the distance from these points to the *medial axis* $\mathrm{ax}(\mathcal{S})$, the set of points in $\mathbb{R}^d$ for which the closest point in $\mathcal{S}$ is not unique. Federer also introduced the reach at a point $p \in \mathcal{S}$ to be the distance from $p$ to the medial axis of $\mathcal{S}$. We now call this quantity the *local feature size* [3] and denote it by $\mathrm{lfs}(p)$.

Federer proved that the reach is stable under $C^{1,1}$ diffeomorphisms of the ambient space. Here, a $C^{1,1}$ map is a $C^1$ map whose derivative is Lipschitz, and a $C^{1,1}$ diffeomorphism is a $C^{1,1}$ bijective map whose inverse is also $C^{1,1}$. Chazal and Soufflet [13] proved that the medial axis is stable with respect to the Hausdorff distance under ambient diffeomorphisms, but under stronger assumptions than the work of Federer, namely assuming that $\mathcal{S}$ is a $C^2$ manifold and the distortion is a $C^2$ diffeomorphism of the ambient space. Chazal and Soufflet based their work on earlier results by Blaschke [9], which were not as strong as Federer's.

In this paper we extend the stability result of the medial axis. More concretely, we generalize the result of Chazal and Soufflet [13] to arbitrary closed sets and $C^{1,1}$ diffeomorphisms of the ambient space; we show that the Hausdorff distance between the medial axes of the closed set and its image is bounded in terms of Lipschitz constants stemming from the diffeomorphism of the ambient space. Our result follows from the work of Federer [19] and in fact shortens the proof in [13] significantly.

Our bounds on the Hausdorff distance say nothing about the topology of the medial axis, which is known to be highly unstable (see e.g. [5]), although it preserves the homotopy type (see [28]).

**Contribution and related work**     Our work differs from the majority of the literature in three essential ways:

Firstly, we make no assumptions on the set we consider apart from that it is closed. The stability of the medial axis of (piecewise) smooth manifolds has been the object of intense study, see for example

Submitted to 37th Conference on Neural Information Processing Systems (NeurIPS 2023). Do not distribute.

[13, 15–17, 24, 30, 37–40]. However, the manifold assumption is impossible to achieve in many applications — such as in the context of astrophysics, one of the main motivations of this paper.

Secondly, we achieve stability without pruning the medial axis. This contrasts with a large body of work, such as [6, 12, 16, 29]. Not having to prune the medial axis is a significant advantage. On the downside, we limit the changes of the considered set to those induced by ambient diffeomorphisms. Nevertheless, given the standard examples of the instability of the medial axis — see for example [5] — we believe these limitations are near to the weakest assumptions necessary for Hausdorff stability. Within the context of ambient homeomorphisms, the results we obtain are close to optimal, as we specify in Remark 4.2.

Thirdly, our results hold for sets in arbitrary dimensions and are not sensitive to the dimension of the set itself. A large part of the related work only investigates sets of low dimensions or codimension one manifolds, although there are some notable exceptions such as [39], see also [17], and [12, 29].

**Motivation**   The medial axis has many real world applications — among others, in robot motion planning [27], triangulation algorithms [4], graphics [35], and shape recognition, segmentation, and learning [10, 18, 25, 33, 41]. See also the overviews [32, 35]. The reach — the distance between a set and its medial axis — is a central concept in manifold learning [1, 2, 20–22, 34].

The motivation of this paper is twofold: Firstly, we tackle the following challenge from  the processing of images collected with optical devices which use lenses — such as cameras or telescopes. A shape extracted from such an image may be imprecise due to the imperfection of the lenses.  Our result implies that the medial axis of such a shape is stable under these imperfections. As a consequence, the outcome of any shape recognition or shape segmentation algorithm based on the medial axis will be stable.

In addition to the disciplines listed above, such stability is sought after in astrophysics, in particular for shape analysis and automated shape identification in observational astronomy.  Observational astronomers are interested in reconstructing objects like stars or galaxies, and their place in the universe from data gathered by telescopes. They can deduce the distance from the object to the observer thanks to so-called standard candles or red shift [14, 23, 31]. However, the image gets distorted due to optical effects — either through gravitational lensing ([7]) or lensing inside the telescope itself ([36]).

Such a distortion can be modeled as a diffeomorphism of the ambient space. At the same time, this problem cannot be tackled using the result by Chazal and Soufflet [13], since the observed objects might not be smooth — for example due to interactions with shock waves or jets. In addition, with our method astrophysicists can not only reconstruct objects in space (3D), but also in spacetime (4D).

The second motivation is more formal in nature: The stability of the medial axis is instrumental in establishing its computability. Indeed, when proving properties of algorithms based on the medial axis, authors generally assume the real RAM model.[1] However, as was recently argued in [29], the medial axis needs to be stable in order to be computable in more realistic models of computation.

There is a more practical component to this formal question: It is not a priori clear if using possibly noisy real world data or the output of other computer programs as input for these algorithms yields answers that are close to the ground truth. To be able to prove that the output is correct, we need (numerical) stability of the medial axis.

**Outline**   After revisiting preliminaries and known results in Section 2, we state the main stability result in Section 3. In Section 4 we reformulate this result in terms of norms on Banach spaces. This also exhibits the fact that the stability of the medial axis is Lipschitz in the following sense: We think of the set $\mathcal{S}$ as fixed and consider the map from the space of diffeomorphisms (endowed with a norm which makes it a Banach space) to the space of closed subsets of $\mathbb{R}^d$ (endowed with the Hausdorff distance), mapping each diffeomorphism $F : \mathbb{R}^d \to \mathbb{R}^d$ to the closure of the medial axis of $F(\mathcal{S})$. The Lipschitz constant then only depends on the diameter of the bounding sphere of the set $\mathcal{S}$.

---

[1]The real RAM model is a standard, albeit non-realistic, assumption in Computational Geometry. It assumes one can calculate precisely with real numbers, instead of using 0s and 1s (which is the usual assumption in computer science).

81 We only include proof sketches of the two main theorems in this article. The full proofs of the
82 theorems and of the supporting lemmas, can be found in the supplementary material.

## 2  Preliminaries: Sets of positive reach and the closest point projection

84 In this section we recall some definitions and results concerning the medial axis and sets of positive
85 reach. Essentially, we need three ingredients from the literature to prove our main theorem: the
86 notions related to the closest point projection, the properties of the generalized normal and tangent
87 spaces, and Federer's result on the stability of the reach under ambient diffeomorphisms.

88 We write $d(\cdot, \cdot)$ for the Euclidean distance between two points, and the distance between a point and
89 a set. That is, for any closed set $\mathcal{S}$ and point $p$,
$$d(p, \mathcal{S}) = \inf_{q \in \mathcal{S}} d(p, q).$$

90 We denote the Hausdorff distance between two sets $A, B \subseteq \mathbb{R}^d$ by $d_H(A, B)$:
$$d_H(A, B) = \max \left\{ \sup_{a \in A} d(a, B), \sup_{b \in B} d(b, A) \right\}.$$

91 We write $B(c, r)$, resp. $S(c, r)$, to denote balls, resp. spheres, with centre $c$ and radius $r$. Lastly, $|\cdot|$
92 denotes the Euclidean norm, and $\|\cdot\|$ an operator norm.

**The closest point projection and related notions**  The projection of points in the ambient space
94 $\mathbb{R}^d$ to the (set of) closest point(s) of the set $\mathcal{S} \subseteq \mathbb{R}^d$ is denoted by $\pi_{\mathcal{S}}$, and illustrated in Figure 1.

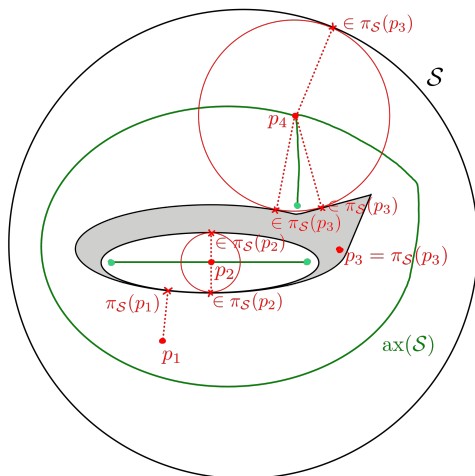

Figure 1: The closest point projection to the set $\mathcal{S}$ of four points in $\mathbb{R}^2$. When a point lies on the medial axis $\mathrm{ax}(\mathcal{S})$, the closest point projection consists of more points.

95 The *medial axis* of $\mathcal{S}$ is the set of all points $p \in \mathbb{R}^d$ where the set $\pi_{\mathcal{S}}(p)$ consists of more than one
96 point:
$$\mathrm{ax}(\mathcal{S}) = \left\{ p \in \mathbb{R}^d \mid \#\pi_{\mathcal{S}}(p) > 1 \right\}.$$

97 Here, $\#\pi_{\mathcal{S}}(p)$ denotes the cardinality of the set $\pi_{\mathcal{S}}(p)$.

98 For a point $p \in \mathcal{S}$, the *local feature size of $p$* is the distance from $p$ to the medial axis of the set $\mathcal{S}$:
$$\mathrm{lfs}(p) = d(p, \mathrm{ax}(\mathcal{S})).$$

99 Finally, the *reach* of the set $\mathcal{S}$ is the infimum of the local feature size over all its points:
$$\mathrm{rch}(\mathcal{S}) = \inf_{p \in \mathcal{S}} \mathrm{lfs}(p) = \inf_{p \in \mathcal{S}} d(p, \mathrm{ax}(\mathcal{S})).$$

100 **Throughout this paper we assume that $\mathcal{S} \subseteq \mathbb{R}^d$ is a closed set.** We shall further assume that the set
101 $\mathcal{S}$ as well as its medial axis are bounded, and that the bounding sphere of $\mathcal{S}$ is contained in $\mathcal{S}$ itself.

102  More specifically, we assume that there exists a closed ball $B$ of positive radius such that $\mathcal{S} \subseteq B$,
103  and $\partial B \subseteq \mathcal{S}$. We call $\partial B$ the bounding sphere of $\mathcal{S}$.

104  The addition of the bounding sphere $\partial B$ to the set $\mathcal{S}$ is necessary to obtain the desired bound on
105  the Hausdorff distance between the two medial axes of the set $\mathcal{S}$ and its image under the ambient
106  diffeomorphism. Indeed, consider the following example, illustrated in Figure 2.

107  Let the set $\mathcal{S}$ consist of two points in the plane, $\mathcal{S} = \{p, q\} \subseteq \mathbb{R}^2$. The medial axis of $\mathcal{S}$ is then the
108  bisector line of $p$ and $q$. After a generic perturbation $F$ of $p$ and $q$ — that is, not a translation and not a
109  perturbation in the direction $\pm(p-q)$ — the bisector line $\mathrm{ax}(F(\mathcal{S}))$ of the perturbed points intersects
110  the bisector $\mathrm{ax}(\mathcal{S})$ of the original pair. The Hausdorff distance between these two non-parallel lines
111  is infinite, and thus unboundable.

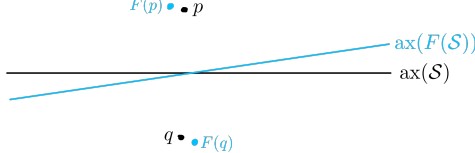

Figure 2: In black the set $\mathcal{S}$ and its medial axis, in light blue the perturbed set and its medial axis. The Hausdorff distance between $\mathrm{ax}(\mathcal{S})$ and $\mathrm{ax}(F(\mathcal{S}))$ is infinite.

112  At the same time, the addition of the bounding sphere $\partial B$ to the considered set $\mathcal{S}$ is not a restriction.
113  Indeed,

114  **Remark 2.1**  *The medial axes of $\mathcal{S}$ and $\mathcal{S} \setminus \partial B$ coincide in the interior of the ball $B$ sufficiently far*
115  *away from its boundary $\partial B$. More precisely:*

116      • *Any point $x \in \mathrm{ax}(\mathcal{S})$, such that $\pi_{\mathcal{S}}(x) \cap \partial B = \emptyset$, lies on the medial axis $\mathrm{ax}(\mathcal{S} \setminus \partial B)$.*
117      • *Conversely, if a point $x$ lies on the medial axis $\mathrm{ax}(\mathcal{S} \setminus \partial B)$, and any (and thus every) point*
118          *$q \in \pi_{\mathcal{S} \setminus \partial B}(x)$ satisfies $d(x, q) < d(x, \partial B)$, then $x \in \mathrm{ax}(\mathcal{S})$.*

119  *Thus, the medial axis is locally stable if the ambient diffeomorphism is close to the identity.*[2]

120  A recurring strategy in this article is to start at a point $p$ on the set $\mathcal{S}$, move away from this point in a
121  'normal' direction, and see if by projecting using the closest point projection $\pi_{\mathcal{S}}$ we get back to $p$. To
122  this end, we define the *projection range*.

123  **Definition 2.2 (Projection range)**  *Let $p \in \mathcal{S}$ be a point and $v \in \mathbb{R}^d$ a vector. The* projection range
124  *$d(p, v, \pi_{\mathcal{S}})$ in direction $v$ is the maximal distance one can travel from $p$ along $v$ such that the closest*
125  *point projection yields only the point $p$:*

$$d(p, v, \pi_{\mathcal{S}}) = \sup\{\lambda \in \mathbb{R} \mid \pi_{\mathcal{S}}(p + \lambda v) = \{p\}\}.$$

126  Since $\pi_{\mathcal{S}}(p) = \{p\}$, the projection range is canonically non-negative. Furthermore, the directions for
127  which the range is positive are key to our study, because of the following property:

128  **Lemma 2.3 (Theorem 4.8 (6) of [19])**  *Consider a point $p \in \mathcal{S}$ and a vector $v \in \mathbb{R}^d$. If*

$$0 < d(p, v, \pi_{\mathcal{S}}) < \infty,$$

129  *then $p + d(p, v, \pi_{\mathcal{S}}) \cdot v \in \overline{\mathrm{ax}(\mathcal{S})}$.*

130  We call these special directions $v$ *back projection vectors*:

131  **Definition 2.4 (Unit back projection vectors)**  *For a point $p \in \mathcal{S}$, $\mathrm{UBP}(p, \mathcal{S})$ is the set of unit*
132  *vectors with a positive projection range:*

$$\mathrm{UBP}(p, \mathcal{S}) = \left\{ u \in \mathbb{R}^d \mid |u| = 1 \text{ and } 0 < d(p, u, \pi_{\mathcal{S}}) < \infty \right\}.$$

133  *We further define*

$$\mathrm{UBP}(\mathcal{S}) = \left\{ (p, u) \in \mathcal{S} \times \mathbb{R}^d \,\middle|\, u \in \mathrm{UBP}(p, \mathcal{S}) \right\},$$
$$\overline{\mathrm{BP}(\mathcal{S})} = \left\{ (p, \lambda u) \in \mathcal{S} \times \mathbb{R}^d \,\middle|\, (p, u) \in \mathrm{UBP}(\mathcal{S}), \lambda \geq 0 \right\}.$$

---

[2]The bounding sphere does allow one to give a relatively clean mathematical statement, see Section 4.

Thanks to Lemma 2.3, the following map is well-defined:

$$\pi_{\mathrm{ax},\mathcal{S}} : \mathrm{UBP}(\mathcal{S}) \to \overline{\mathrm{ax}(\mathcal{S})}, \qquad (p, u) \mapsto p + d(p, u, \pi_{\mathcal{S}})u. \qquad (1)$$

**The generalized tangent and normal space**  Back projection vectors are intricately related to the generalized tangent and normal spaces.

**Definition 2.5 (Definitions 4.3 and 4.4 of [19])** *Let $p \in \mathcal{S}$. The generalized tangent space $\mathrm{Tan}(p, \mathcal{S})$ is the set of vectors $u \in \mathbb{R}^d$, such that either $u = 0$ or, for every $\varepsilon > 0$ there exists a point $q \in \mathcal{S}$ with*

$$0 < |q - p| < \varepsilon \qquad\qquad and \qquad\qquad \left| \frac{q - p}{|q - p|} - \frac{u}{|u|} \right| < \varepsilon.$$

*The generalized normal space $\mathrm{Nor}(p, \mathcal{S})$ consists of vectors $v \in \mathbb{R}^d$ such that $\langle v, u \rangle \leq 0$ for all $u \in \mathrm{Tan}(p, \mathcal{S})$. Vectors contained in the generalized tangent, resp. normal, space are called* tangent, *resp.* normal, *to $\mathcal{S}$ at $p$.*

The generalized tangent and normal spaces are illustrated in Figure 3.

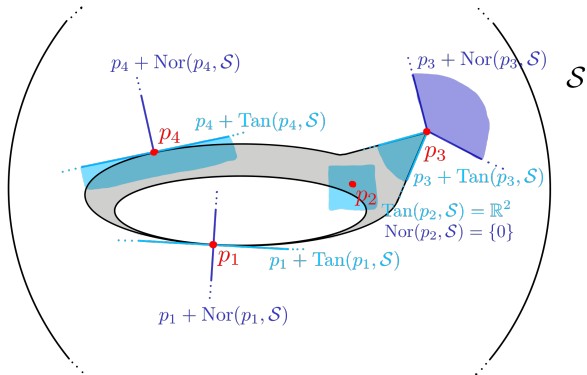

Figure 3: The (affine) generalized tangent and normal spaces of four points in the set $\mathcal{S} \subset \mathbb{R}^2$, in light blue and violet, respectively.

**Stability of the reach under ambient diffeomorphisms**  Our last ingredient is the following result by Federer.

**Theorem 2.6 (Stability of the reach under ambient diffeomorphisms, Theorem 4.19 of [19])**
*Pick two constants $0 < t < \mathrm{rch}(\mathcal{S})$ and $s > 0$. If the map*

$$F : \{x \in \mathbb{R}^d \mid d(x, \mathcal{S}) < s\} \to \mathbb{R}^n$$

*is injective and continuously differentiable, and the maps $F$, $F^{-1}$, and $DF$ are Lipschitz continuous with Lipschitz constants $\mathrm{Lip}(F)$, $\mathrm{Lip}(F^{-1})$, $\mathrm{Lip}(DF)$, respectively, then the reach $\mathrm{rch}(F(\mathcal{S}))$ of the image of the set $\mathcal{S}$ under the map $F$ is lower-bounded by*

$$\mathrm{rch}(F(\mathcal{S})) \geq \min \left\{ \frac{s}{\mathrm{Lip}(F^{-1})}, \frac{1}{\left( \frac{\mathrm{Lip}(F)}{t} + \mathrm{Lip}(DF) \right) (\mathrm{Lip}(F^{-1}))^2} \right\}.$$

## 3  Stability of the medial axis under ambient diffeomorphisms

In this section we present the main result of this paper, Theorem 3.9. This theorem extends earlier work by Chazal and Soufflet [13]. Its proof relies on Federer's result on the stability of the reach, Theorem 2.6. To give a more geometrical interpretation we introduce the concept of a weakly tangent sphere and ball, and a maximal empty weakly tangent ball.

**Definition 3.1 (Weakly tangent sphere and ball)** *Let $p \in \mathcal{S}$. A sphere is called* weakly tangent *to $\mathcal{S}$ at $p$ if it contains the point $p$ and its centre lies in the (translated) generalized normal space $\mathrm{Nor}(p, \mathcal{S}) + p$. In other words, spheres weakly tangent to $\mathcal{S}$ at $p$ are spheres with centres $p + v$ and radii $|v|$, for a vector $v \in \mathrm{Nor}(p, \mathcal{S})$.*

*A ball is called* weakly tangent *to $\mathcal{S}$ at $p$ if its boundary sphere is* weakly tangent *to $\mathcal{S}$ at $p$.*

**Remark 3.2** *Using the definition of $\mathrm{Nor}(p, \mathcal{S})$, a weakly tangent ball can also be defined as follows: A ball $B(c, r)$ is weakly tangent at $p$ if and only if its centre $c$ and radius $r$ satisfy*

$$(p + \mathrm{Tan}(\mathcal{S}, p)) \cap B(c, r) = \{p\}.$$

We remark:

**Lemma 3.3** *Let $p \in \mathcal{S}$ and $v \in \mathbb{R}^d$, and suppose that for some $\lambda > 0$ we have $\pi_{\mathcal{S}}(p + \lambda v) \neq \{p\}$. Then, for all $\lambda' \geq \lambda$, we have $\pi_{\mathcal{S}}(p + \lambda' v) \neq \{p\}$ and for all $\lambda' > \lambda$, that $p \notin \pi_{\mathcal{S}}(p + \lambda' v)$.*

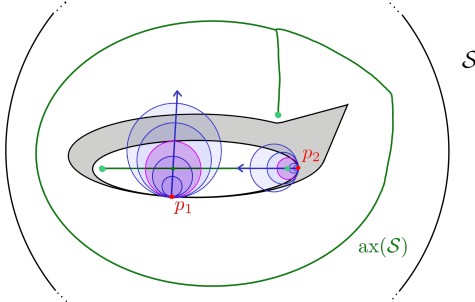

Figure 4: Two families of balls weakly tangent to the set $\mathcal{S} \subset \mathbb{R}^2$ (in blue). Each family contains a unique maximal empty ball (in purple). Notice that the centre of the maximal empty ball weakly tangent at the point $p_1$ lies at the medial axis $\mathrm{ax}(\mathcal{S})$, while the centre of the maximal empty ball weakly tangent at the point $p_2$ only lies at its closure, $\overline{\mathrm{ax}(\mathcal{S})}$.

Lemma 3.3 essentially tells us that a family of weakly tangent balls $\{B(p + \lambda v, \lambda|v|)\}_{\lambda \geq 0}$ contains at most one which is maximal with respect to inclusion among those whose interior is disjoint from the set $\mathcal{S}$. Two such families are illustrated in Figure 4.

We call such balls *maximal empty*. For the purpose of this article, we define maximal empty balls in terms of unit back projection vectors (Definition 2.4). To see that each maximal empty ball is indeed weakly tangent, we emphasise:

**Lemma 3.4** *If $(p, v) \in \mathrm{BP}(\mathcal{S})$, then $(p, v) \in \mathrm{Nor}(\mathcal{S})$. That is, $\mathrm{BP}(\mathcal{S}) \subseteq \mathrm{Nor}(\mathcal{S})$. In particular, for any pair $(p, u) \in \mathrm{UBP}(\mathcal{S})$ and radius $\lambda \geq 0$, the ball $B(p + \lambda u, \lambda)$ is weakly tangent to $\mathcal{S}$.*

**Remark 3.5** *For general closed sets, the converse of Lemma 3.4, that is, $\mathrm{Nor}(\mathcal{S}) \subseteq \mathrm{BP}(\mathcal{S})$, is not true. One counter-example is the graph of the function $x \mapsto |x|^{3/2}$ at the origin. However, the inclusion $\mathrm{Nor}(\mathcal{S}) \subseteq \mathrm{BP}(\mathcal{S})$ holds for sets of positive reach, thanks to Theorem 4.8 (12) of [19] (recalled in the supplementary material).*

**Definition 3.6 (Maximal empty weakly tangent ball)** *Let $(p, u) \in \mathrm{UBP}(\mathcal{S})$. A weakly tangent ball $B(p + \lambda u, \lambda)$ is called* maximal empty *to $\mathcal{S}$ if $\lambda = d(p, u, \pi_{\mathcal{S}})$, or, equivalently, if $\pi_{\mathrm{ax}, \mathcal{S}}(p, u) = p + \lambda u$.*

(Maximal empty) weakly tangent balls satisfy the following properties. Let $(p, u) \in \mathrm{UBP}(\mathcal{S})$.

- For any radius $0 < \lambda \leq d(p, u, \pi_{\mathcal{S}})$, the interior of the ball $B(p + \lambda u, \lambda)$ is disjoint from the set $\mathcal{S}$. This follows directly from Definition 3.6 and Lemma 3.3.

- The centres of maximal empty weakly tangent balls lie on the closure of the medial axis of $\mathcal{S}$. This is due to Lemma 2.3 and the definition of the map $\pi_{\mathrm{ax}, \mathcal{S}}$ (equation (1)).

186 The following lemma moreover tells us, that each point on the medial axis is a centre of a maximal
187 empty weakly tangent ball.

188 **Lemma 3.7 (Surjectivity on** $\mathrm{ax}(\mathcal{S})$**)** *For any point $x \in \mathrm{ax}(\mathcal{S})$ and $p \in \pi_{\mathcal{S}}(x)$, there exists a vector*
189 $u \in \mathrm{UBP}(p, \mathcal{S})$ *such that $\pi_{\mathrm{ax},\mathcal{S}}(p, u) = x$. In other words, $B(x, |x - p|)$ is a maximally empty*
190 *weakly tangent ball. Moreover, we have that*

$$\mathrm{ax}(\mathcal{S}) \subseteq \pi_{\mathrm{ax},\mathcal{S}}\left(\mathrm{UBP}(\mathcal{S})\right) \subseteq \overline{\mathrm{ax}(\mathcal{S})}.$$

191 We are now almost ready to state our main theorem. Before phrasing the result, we walk the reader
through the assumptions and fix the notation on the way. The assumptions are illustrated in Figure 5.

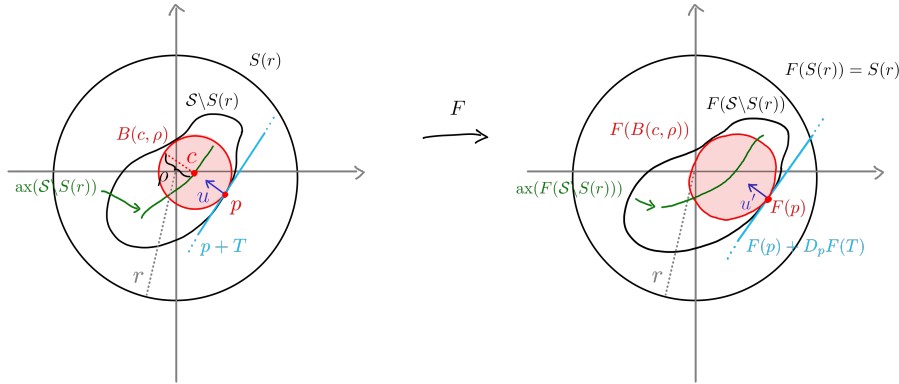

Figure 5: The setting of Theorem 3.9.

192

193 **Assumption 3.8**

194 - *We assume that the set $\mathcal{S}$ has a bounding sphere of radius $r$, which we denote by $S(r)$.*
195 - *We consider a $C^1$ diffeomorphism $F : \mathbb{R}^d \to \mathbb{R}^d$ such that the Lipschitz constants of $F$ and*
196 $F^{-1}$ *are bounded by $L_F$, and the Lipschitz constants of the differentials $DF$ and $DF^{-1}$*
197 *are bounded by $L_{DF}$. We call such a diffeomorphism a $C^{1,1}$ diffeomorphism.*
198 - *We further assume that the map $F$ leaves the bounding sphere $S(r)$ invariant, that is,*
199 $F(S(r)) = S(r)$.
200 - *We pick a point $c \in \mathrm{ax}(\mathcal{S})$, a point $p \in \pi_{\mathcal{S}}(c)$, and write $\rho = |c - p|$. Observe that since*
201 $\mathcal{S} \cap \mathrm{ax}(\mathcal{S}) = \emptyset$, $\rho$ *is positive. By Lemma 3.7, the ball $B(c, \rho)$ is a maximal empty weakly*
202 *tangent ball to $\mathcal{S}$ at $p$. Moreover, we define $u = \frac{c-p}{|c-p|}$ and note that $u \in \mathrm{UBP}(p, \mathcal{S})$.*
203 - *We denote the tangent hyperplane to the boundary sphere of $B(c, \rho)$ at $p$ by $p + T$. The*
204 *hyperplane $T$ is the orthocomplement of the vector $u$: $T = u^\perp$.*
205 - *We work with the unit vector at $F(p)$ that points inside the image of the ball $B(c, \rho)$ and is*
206 *orthogonal to the hyperplane $D_p F(T)$. We denote this vector by $u'$.*

207 **Theorem 3.9** *Under the above assumptions, there exists a maximal empty weakly tangent ball*
208 $B(c', \rho')$ *to the set $F(\mathcal{S})$, whose boundary sphere has an internal normal $u'$. In particular, the*
209 *ball $B(c', \rho')$ is tangent to the affine hyperplane $F(p) + D_p F(T)$. Its radius $\rho'$ is bounded by*
210 $\rho' \in \left[ \frac{\rho}{(L_F)^3 + \rho L_{DF}(L_F)^2}, \frac{(L_F)^3 \rho}{1 - \rho L_{DF}(L_F)^2} \right]$ *. Assume, moreover, that the distortions of both $F$ and $DF$*
211 *are bounded, that is, for all $x \in \mathbb{R}^d$,*

$$|F(x) - x| \leq \varepsilon_1, \qquad \|DF_x - \mathrm{Id}\| \leq \varepsilon_2 < 1, \tag{2}$$

212 *and $r \cdot L_{DF}(L_F)^2 \leq 1/2$. Define*

$$C_L(r, L_F, L_{DF}, \varepsilon_1, \varepsilon_2) =$$

$$2r\sqrt{1 + (L_F)^6 \left(1 + 4r L_{DF}(L_F)^2\right)^2 - 2(L_F)^3 \left(1 + 4r L_{DF}(L_F)^2\right) \sqrt{1 - (\varepsilon_2)^2}} + \varepsilon_1$$

213 *then the map $\pi_{\mathrm{ax},\mathcal{S}}$ satisfies*

$$|\pi_{\mathrm{ax},\mathcal{S}}(p, u) - \pi_{\mathrm{ax}, F(\mathcal{S})}(F(p), u')| \leq C_L(r, L_F, L_{DF}, \varepsilon_1, \varepsilon_2).$$

*Thus, the Hausdorff distance between the medial axes of $\mathcal{S}$ and its image $F(\mathcal{S})$ is bounded by*

$$d_H(\mathrm{ax}(\mathcal{S}), \mathrm{ax}(F(\mathcal{S}))) \leq C_L(r, L_F, L_{DF}, \varepsilon_1, \varepsilon_2). \tag{3}$$

The bound $|F(x) - x| \leq \varepsilon_1$ is really necessary, because we want our theorem to accommodate for rotations and translations, which rotate and translate the medial axis without changing distances and hence have Lipschitz constant 1. We further stress that if the diffeomorphism $F$ is close to the identity, its Lipschitz constant satisfies $L_F \geq 1$, because by assumption $F$ leaves the bounding sphere $S(r)$ invariant, and $L_{DF}$ is close to zero.

*Sketch of the proof of Theorem 3.9*    The idea of the proof is depicted in Figure 5. Thanks to Federer's result (Theorem 2.6), we know that the reach of the maximal empty weakly tangent ball $B(c, \rho)$ does not change too much under the ambient diffeomorphism $F$. This gives a lower bound on the radius of every maximal empty weakly tangent ball of the image of this ball — the set $F(B(c, \rho))$. We show that in the interior of $F(B(c, \rho))$, the radii of the maximal empty weakly tangent balls of $F(B(c, \rho))$ are close to $\rho$. One of these balls is also empty weakly tangent to $F(\mathcal{S})$ at $F(p)$, though not necessarily maximal. We denote its centre by $c'$. Since we can apply the same argument for the map $F^{-1}$, we find an upper and lower bound on the radius of the maximal weakly tangent ball $B(c', \rho')$ of $F(\mathcal{S})$ at $F(p)$ that is also weakly tangent to $F(B(c, \rho))$, or equivalently tangent to $D_p F(T)$.

While this bound on the difference of the radii is essentially a bound on the distance $||c - p| - |c' - F(p)||$ between the points $c - p$ and $c' - F(p)$, the bound $\varepsilon_2$ on $\|DF - 1\|$ allows one to bound the angle between the vectors $c - p$ and $c' - F(p)$. With the assumption (2) we can then derive a bound the distance between the points $c$ and $c'$. Finally, thanks to [19, Theorem 4.8 (6)] (Lemma 2.3) this induces a bound on the Hausdorff distance between the (closure of the) two medial axes $\mathrm{ax}(S)$ and $\mathrm{ax}(F(S))$. $\qquad\square$

It was a surprise to the authors that no assumption on the set (apart from closedness) needed to be made, and that the techniques used were that simple and well established; they go back to Federer [19]. In fact, the authors at first envisioned a far more elaborate argument assuming the set had positive $\mu$-reach [11].

# 4    Quantifying $C^{1,1}$ diffeomorphisms as deviations from identity

In this section we reformulate the main result in terms of norms on Banach spaces. This reformulation offers a more theoretical insight, and we believe the reformulated bounds are easier to work with in certain applications.    Indeed, in the context of practical numerical computations, a bound on the Lipschitz constant of an operator — or, at least, a modulus of continuity — allows to control the condition number. This control is particularly useful when we calculate with objects such as the medial axis, whose (numerical) stability is often problematic in practice.

As we will see below, for this reformulation we somewhat strengthen our assumptions.

We decompose a diffeomorphism $F$ into the identity map $\mathbb{1}_{\mathbb{R}^d}$ on $\mathbb{R}^d$, and a displacement field $\varphi$: $F = \mathbb{1}_{\mathbb{R}^d} + \varphi$. For the choice of the displacement field, we restrict ourselves to the vector space $\mathcal{U}$ of all $C^{1,1}$ maps $\varphi$ from $\mathbb{R}^d$ to $\mathbb{R}^d$ whose restriction to the exterior $\mathbb{R}^d \setminus B(r)$ of a certain bounding ball $B(r)$ equals 0.[3]

A natural norm associated to $\mathcal{U}$ is one that makes it a Banach space. A typical choice, inherited from general Banach spaces of $C^{1,1}$ functions, would be for example, for $\varphi \in \mathcal{U}$,

$$\|\varphi\|_{C^{1,1}} = \max\left(\|\varphi\|_\infty, \|D\varphi\|_\infty, \mathrm{Lip}(D\varphi)\right). \tag{4}$$

Here we used the following notation:

- $\|\varphi\|_\infty = \sup_{x \in \mathbb{R}^d} |\varphi(x)|$ denotes the sup norm on $x \mapsto |\varphi(x)|$, where $|\cdot|$ is the Euclidean norm in $\mathbb{R}^d$,
- $\|D\varphi\|_\infty = \sup_{x \in \mathbb{R}^d} \|D\varphi(x)\|$ denotes the sup norm on $x \mapsto \|D\varphi(x)\|$, where $\|D\varphi(x)\|$ is the operator norm induced by the Euclidean norm on $\mathbb{R}^d$.

---

[3]This is more restrictive than assuming that the restriction to the bounding sphere $S(r)$ is 0, but it simplifies matters in this section.

- We write $\mathrm{Lip}(D\varphi)$ for the Lipschitz semi-norm of $D\varphi$. The Lipschitz semi-norms of $\varphi$ and $D\varphi$ are defined as

$$\mathrm{Lip}(\varphi) = \sup_{x,y \in \mathbb{R}^d,\, x \neq y} \frac{|\varphi(y) - \varphi(x)|}{|y - x|},$$

and

$$\mathrm{Lip}(D\varphi) = \sup_{x,y \in \mathbb{R}^d,\, x \neq y} \frac{\|D\varphi(y) - D\varphi(x)\|}{|y - x|}.$$

The norm defined in (4) makes $\mathcal{U}$ into a Banach space, since every Cauchy sequence in $\mathcal{U}$ has a limit in $\mathcal{U}$. In addition, any function $\varphi \in \mathcal{U}$ satisfies:

$$\mathrm{Lip}(\varphi) = \|D\varphi\|_\infty, \tag{5}$$

$$\|D\varphi\|_\infty \leq r \,\mathrm{Lip}(D\varphi), \tag{6}$$

$$\|\varphi\|_\infty \leq r \,\mathrm{Lip}(\varphi) \leq r^2 \,\mathrm{Lip}(D\varphi), \tag{7}$$

since the restriction of $\varphi$ to $\mathbb{R}^d \setminus B(r)$ is 0. This in turn yields that $\mathrm{Lip}(D\varphi) \leq \|\varphi\|_{C^{1,1}} \leq \max(1, r, r^2)\,\mathrm{Lip}(D\varphi)$. Thus, in $\mathcal{U}$, the norm $\varphi \mapsto \mathrm{Lip}(D\varphi)$ is equivalent to the norm $\varphi \mapsto \|\varphi\|_{C^{1,1}}$.

We can now state slightly less general version of Theorem 3.9 in terms of the Banach space $(\mathcal{U},\, \varphi \mapsto \mathrm{Lip}(D\varphi))$.

**Theorem 4.1** *Let $\mathcal{S} \subseteq \mathbb{R}^d$ be bounded by the ball $B(r)$ of radius $r > 0$, such that $S(r) = \partial B(r) \subseteq \mathcal{S}$. Let further $F$ be a $C^{1,1}$ diffeomorphism from $\mathbb{R}^d$ to itself that leaves the set $\mathbb{R}^d \setminus B(r)$ invariant, and define two displacement fields $\varphi, \tilde{\varphi} \in \mathcal{U}$ such that $F = \mathbb{1}_{\mathbb{R}^d} + \varphi$ and*

$$(\mathbb{1}_{\mathbb{R}^d} + \tilde{\varphi}) \circ (\mathbb{1}_{\mathbb{R}^d} + \varphi) = \mathbb{1}_{\mathbb{R}^d}.$$

*Define $\varepsilon = \max\left(\mathrm{Lip}(D\varphi), \mathrm{Lip}(D\tilde{\varphi})\right)$.*

*If $r\varepsilon \leq 1/4$ , the Hausdorff distance between the medial axes of the set $\mathcal{S}$ and its image $F(\mathcal{S})$ is bounded by $d_H(\mathrm{ax}(\mathcal{S}), \mathrm{ax}(F(\mathcal{S}))) \leq \left(1 + \sqrt{50}\right) r^2 \varepsilon + \mathcal{O}\left(r^3 \varepsilon^2\right)$. In particular, $d_H(\mathrm{ax}(\mathcal{S}), \mathrm{ax}(F(\mathcal{S}))) = \mathcal{O}\left(r^2 \varepsilon\right)$.*

*Sketch of the proof*    Essentially, the proof consists of rewriting Theorem 3.9 in terms of the language developed in this section.  □

**Remark 4.2** *Observe that the bound $\mathcal{O}\left(r^2 \varepsilon\right)$ is consistent with a scaling by factor $\lambda$: $\mathcal{S} \mapsto \lambda \mathcal{S}$, $F(\cdot) \mapsto \lambda F(\cdot/\lambda)$. Under such a scaling, the radius $r$ is multiplied by $\lambda$, while the Lipschitz constant $\mathrm{Lip}(D\varphi)$ — and therefore $\varepsilon$ — is divided by $\lambda$. Furthermore, the Hausdorff distance $d_H(\mathrm{ax}(\mathcal{S}), \mathrm{ax}(F(\mathcal{S})))$ increases by a factor $\lambda$. By considering a diffeomorphism that translates the set $\mathcal{S} \setminus S(r)$ while keeping the bounding sphere $S(r)$ fixed, we see that this bound is asymptotically optimal.*

# 5 Conclusion and future work

We proved the Hausdorff stability of the medial axis of a closed set without any further assumption on it (as explained in Remark 2.1, the existence of the bounding sphere serves to formulate the main result in a clean way).

With regard to applications, our result is the first step towards providing a provably correct image recognition in particular in the context of astrophysics. The next step is to produce physics-informed models for the medial axis as occurring in astronomical data.

On the mathematical side, we conclude with a conjecture generalizing our result. We believe that our result generalizes to compact Riemannian manifolds with bounded curvature.

**Conjecture 5.1** *Let $\mathcal{M}$ be a compact Riemannian manifold with bounded sectional curvature[4] and $\mathcal{S}$ a closed subset of $\mathcal{M}$. Then the medial axis (also called cut locus [26]) of $\mathcal{S}$ in $\mathcal{M}$ is Lipschitz stable under diffeomorphisms of $\mathcal{M}$.*

---

[4]See [8] for definitions and a very pedagogical overview of the properties of these manifolds.

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
