# Supplementary material to the article: The medial axis of closed bounded sets is Lipschitz stable with respect to the Hausdorff distance under ambient diffeomorphisms

## A    Proofs

*Proof of Lemma 3.3*    We first note that the statement is empty if $v = 0$. Let $v \neq 0$, and consider the nested family of balls $B(p + \lambda'v, \lambda'|v|)$, parametrized by $\lambda' > 0$.

Because $\pi_\mathcal{S}(p + \lambda v) \neq p$, the (closed) ball $B(p + \lambda v, \lambda|v|)$ contains a point $q \in \mathcal{S}$ other than $p$. Since the balls $B(p + \lambda'v, \lambda'|v|)$ are nested, the point $q$ lies inside every ball $B(p + \lambda'v, \lambda'|v|)$ with $\lambda' \geq \lambda$. Moreover, $q$ lies in the interior of $B(p + \lambda'v, \lambda'|v|)$ for $\lambda' > \lambda$. Hence, for every $\lambda' \geq \lambda$, we have that $\pi_\mathcal{S}(p + \lambda'v) \neq \{p\}$ and for $\lambda' > \lambda$, that $p \notin \pi_\mathcal{S}(p + \lambda'v)$. $\qquad\square$

*Proof of Lemma 3.7*    Let $Q = \pi_\mathcal{S}(x)$ be the subset of $\mathcal{S}$ that is closest to $x$. Because $x \in \mathrm{ax}(\mathcal{S})$, $Q$ contains at least two points, one of them being $p$. We write $\lambda = |x - p|$. Since $\mathcal{S}$ and $\mathrm{ax}(\mathcal{S})$ are disjoint, $\lambda > 0$, and thus we can define $u = \frac{x - p}{\lambda}$.

Since the interior of the ball $B(x, \lambda)$ does not intersect $\mathcal{S}$, it in particular does not intersect $\mathrm{Tan}(p, \mathcal{S})$ and thus $B(x, \lambda)$ is weakly tangent at $p$ by Remark 3.2. Let us now consider the nested family $B(p + \lambda'u, \lambda')$ of weakly tangent balls at $p$. By definition, $\partial B(x, \lambda) \cap \mathcal{S} = Q$ and therefore $B(p + \lambda'u, \lambda') \cap \mathcal{S} = p$ for $\lambda' < \lambda$. At the same time, Lemma 3.3 yields that for $\lambda' > \lambda$, $p \notin \pi_\mathcal{S}(p + \lambda'u)$. Hence the projection range in direction $u$ equals $d(p, u, \pi_\mathcal{S}) = \lambda$ and we obtain $\pi_{\mathrm{ax},\mathcal{S}}(p, u) = p + \lambda u = x$ directly from Definition 3.6 . The fact that $\pi_{\mathrm{ax},\mathcal{S}}\left(\mathrm{UBP}(\mathcal{S})\right) \subseteq \overline{\mathrm{ax}(\mathcal{S})}$ is due to Lemma 2.3 $\qquad\square$

The next two claims are used in the proof of Theorem 3.9:

**Claim A.1**

$$u' = \frac{(D_pF^t)^{-1}(u)}{|(D_pF^t)^{-1}(u)|}, \tag{8}$$

*where $D_pF^t$ is the transpose matrix (or the adjoint operator) of $DF$ at the point $p$, defined by*

$$\forall v_1, v_2, \qquad \langle v_1, D_pF(v_2)\rangle = \langle D_pF^t(v_1), v_2\rangle.$$

*Proof*

$$\begin{aligned}
w \in D_pF(u^\perp) &\iff \langle D_pF^{-1}(w), u\rangle = 0 \\
&\iff \langle w, (D_pF^{-1})^t u\rangle = 0 \\
&\iff \langle w, u'\rangle = 0 \\
&\iff w \in u'^\perp,
\end{aligned}$$

21 and thus

$$D_pF(u^\perp) = u'^\perp. \tag{9}$$

22 In other words, we have shown that $u'$ is orthogonal to $D_pF(u^\perp) = D_pF(T)$.

23 Because

$$\langle D_pF(u), (D_pF^{-1})^t(u) \rangle = \langle D_pF^{-1}(D_pF(u)), u \rangle = \langle u, u \rangle > 0,$$

24 we deduce that $\langle D_pF(u), u' \rangle > 0$. This is in turn equivalent to $u'$ pointing towards the interior of
25 $F(B(c,\rho))$. $\qquad\square$

26 **Claim A.2** *Let $\|D_pF - \mathrm{Id}\| \leq \varepsilon < 1$. Then the angle $\angle u, u'$ between the vectors $u$ and $u'$ satisfies*

$$\cos \angle u, u' \geq \sqrt{1 - \varepsilon^2}.$$

27 *Proof* We first show that $\angle u, u' < \pi/2$. Indeed, define the vector $w$ as

$$w = (D_pF^t)^{-1}(u),$$

28 that is, the vector satisfying $u = D_pF^t(w)$. Then $u' = \frac{w}{|w|}$ (see equation (8) ), and

$$
\begin{aligned}
|w|\langle u, u' \rangle = \langle u, w \rangle &= \langle D_pF^t w, w \rangle \\
&= \langle w, D_pF w \rangle = |w|^2 + \langle w, (D_pF - \mathrm{Id})w \rangle \\
&\geq |w|^2 - |w|^2 \|D_pF - \mathrm{Id}\| \\
&> 0. \qquad \text{(because, by assumption, } \|DF_p - \mathrm{Id}\| < 1)
\end{aligned}
$$

29 Thus, $\langle u, u' \rangle > 0$, and therefore $\angle u, u' < \pi/2$.

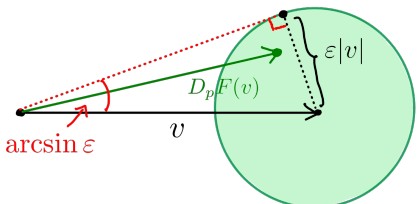

Figure 6: Since $\|v - D_pF(v)\| \leq \varepsilon |v|$, the vector $D_pF(v)$ lies in the green ball $B(v, \varepsilon |v|)$. Since $\varepsilon < 1$, the angle between $v$ and $D_pF(v)$ is upper-bounded by $\arcsin \varepsilon < \pi/2$.

30 Furthermore, consider a vector $v \in u^\perp$. Since $\|v - D_pF(v)\| \leq \|D_pF - \mathrm{Id}\||v| \leq \varepsilon|v|$, the angle
31 between $v$ and $D_pF(v)$ is upper-bounded by $\arcsin \varepsilon < \pi/2$, as illustrated in Figure 6. This yields a
32 bound on the angle between the tangent spaces $u^\perp$ and $D_pF(u^\perp)$:

$$\sin \angle u^\perp, D_pF(u^\perp) = \sin \sup_{v \in u^\perp, w \in D_pF(u^\perp)} \angle v, w \leq \varepsilon. \tag{10}$$

33 Using (9) and (10) we deduce that:

$$\sin \angle u, u' = \sin \angle u^\perp, u'^\perp \leq \varepsilon.$$

34 Finally, since $\angle u, u' < \pi/2$, $\cos \angle u, u' \geq \sqrt{1 - \varepsilon^2}$. This concludes the proof. $\qquad\square$

35 *Proof of Theorem 3.9* We first derive the bounds for the radius $\rho'$. As the first step, we apply
36 Theorem 2.6 to the boundary sphere $S(c,\rho)$ of the maximal empty weakly tangent ball $B(c,\rho)$. In
37 particular, we can choose the constant $s$ in Theorem 2.6 arbitrarily large, and the constant $t$ arbitrarily
38 close to the reach $\mathrm{rch}(S(c,\rho)) = \rho$, to obtain:

$$\mathrm{rch}\left(F(S(c,\rho))\right) \geq \frac{1}{\left(\frac{L_F}{\rho} + L_{DF}\right)(L_F)^2} = \frac{\rho}{(L_F)^3 + \rho L_{DF}(L_F)^2} =: \rho_1.$$

39 This means that no open ball of radius $\rho_1$ tangent to the set $F(S(c,\rho))$ actually intersects $F(S(c,\rho))$.
40 In addition, since the set $F(B(c,\rho))$ does not contain any points of $F(\mathcal{S})$ in its interior, no ball of

41  radius $\rho_1$ that is tangent to $F(S(c, \rho))$ and whose centre lies inside $F(S(c, \rho))$ contains any point of
42  $F(\mathcal{S})$.

43  The unit vector $u' \in DF_p(T)^\perp$ (defined in (8)) is defined such that the point $F(p) + \rho_1 u'$ lies inside
44  the distorted ball $F(B(c, \rho))$. Due to the above observation, the ball $B(F(p) + \rho_1 u', \rho_1)$ is weakly
45  tangent to $F(\mathcal{S})$ at $F(p)$ and contains no points of $F(\mathcal{S})$ in its interior.

46  Let us now consider the weakly tangent ball $B(F(p) + \rho'' u', \rho'')$, whose radius $\rho''$ satisfies

$$\rho'' > \frac{(L_F)^3 \rho}{1 - \rho L_{DF}(L_F)^2} =: \rho_2.$$

47  To shorten up the notation, we set

$$F(p) + \rho'' u' =: c''.$$

48  To derive a contradiction, we assume that $B(c'', \rho'')$ is maximal empty. This is equivalent to
49  assuming that $\mathrm{int} B(c'', \rho'') \cap F(\mathcal{S}) = \emptyset$, and thus $B(c'', \rho'')$ is a maximal empty weakly tangent ball
50  to $F(p)$. Similarly to the beginning of the proof, we now apply Theorem 2.6 to the map $F^{-1}$ and the
51  boundary sphere $S(c'', \rho'') = \partial B(c'', \rho'')$. As a result, the reach of $F^{-1}(S(c'', \rho''))$ is at least

$$\mathrm{rch}\left(F^{-1}(S(c'', \rho''))\right) \geq \frac{\frac{(L_F)^3 \rho}{1 - \rho L_{DF}(L_F)^2}}{(L_F)^3 + \frac{(L_F)^3 \rho}{1 - \rho L_{DF}(L_F)^2} L_{DF}(L_F)^2} = \rho.$$

52  We conclude that there exists a ball that is tangent to the set $F^{-1}(S(c'', \rho''))$ at $F^{-1}(F(p)) = p$,
53  whose radius is larger than $\rho$, and that does not contain any points of $\mathcal{S}$ in its interior. This contradicts
54  the fact that the ball $B(c, \rho)$ is maximal empty, and completes the proof of the first part of the
55  statement.

56  We now prove the bounds on the distortion of the map $\pi_{\mathrm{ax}, \mathcal{S}}$. Let $\rho' \in [\rho_1, \rho_2]$ be the radius of the
57  maximal empty weakly tangent ball at $F(p)$ in the direction $u'$, and write $c' := F(p) + \rho' u'$ for its
58  centre. We stress that, as a consequence of Lemma 2.3, $c' \in \overline{\mathrm{ax}(F(\mathcal{S}))}$, but it is not necessarily true
59  that $c' \in \mathrm{ax}(F(\mathcal{S}))$.

60  The goal is to estimate the distance between the two centres $c = \pi_{\mathrm{ax}, \mathcal{S}}(p, u)$ and $c' = \pi_{\mathrm{ax}, \mathcal{S}}(F(p), u')$.
61  Indeed, since $c - p = \rho u$ and $c' - F(p) = \rho' u'$,

$$|c - c'| = |c - p + p - F(p) + F(p) - c'| = |\rho u + p - F(p) - \rho' c'|$$
$$\leq |\rho u - \rho' u'| + |F(p) - p|.$$

62  Due to the assumptions of the theorem, $|F(p) - p| \leq \varepsilon_1$. Furthermore, thanks to Claim A.2,

$$|\rho u - \rho' u'|^2 = \rho^2 + (\rho')^2 - 2\rho\rho' \cos \angle u, u' \leq \rho^2 + (\rho')^2 - 2\rho\rho' \sqrt{1 - (\varepsilon_2)^2}.$$

63  Recalling that $\rho' \in [\rho_1, \rho_2]$, we thus obtain

$$|\rho u - \rho' u'| \leq \max\left(\sqrt{\rho^2 + (\rho_1)^2 - 2\rho\,\rho_1 \cos(\arcsin(\varepsilon_2))}, \sqrt{\rho^2 + (\rho_2)^2 - 2\rho\,\rho_2 \cos(\arcsin(\varepsilon_2))}\right)$$
$$= \max\left(\sqrt{\rho^2 + (\rho_1)^2 - 2\rho\,\rho_1 \sqrt{1 - (\varepsilon_2)^2}}, \sqrt{\rho^2 + (\rho_2)^2 - 2\rho\,\rho_2 \sqrt{1 - (\varepsilon_2)^2}}\right).$$

64  Hence,

$$|c - c'| \leq \max\left(\sqrt{\rho^2 + (\rho_1)^2 - 2\rho\,\rho_1 \sqrt{1 - (\varepsilon_2)^2}}, \sqrt{\rho^2 + (\rho_2)^2 - 2\rho\,\rho_2 \sqrt{1 - (\varepsilon_2)^2}}\right) + \varepsilon_1.$$
$$\tag{11}$$

65  As the last step, we simplify the expression (11) (at the cost of weakening the bounds). For this, we
66  assume that $\rho L_{DF}(L_F)^2 \leq 1/2$, so that

$$\rho_1 = \frac{\rho}{(L_F)^3 + \rho L_{DF}(L_F)^2} \geq \frac{\rho}{(L_F)^3}\left(1 - \rho \frac{L_{DF}}{L_F}\right), \tag{12}$$

$$\rho_2 = \frac{(L_F)^3 \rho}{1 - \rho L_{DF}(L_F)^2} \leq \rho(L_F)^3\left(1 + 2\rho L_{DF}(L_F)^2\right), \tag{13}$$

67 where we used that, for $x \in [0, 1/2]$, $\frac{1}{1+x} \geq 1 - x$ and $\frac{1}{1-x} \leq 1 + 2x$. We note that both $\rho_1$ and $\rho_2$
68 tend to $\rho$ as $L_F$ tends to 1 and $L_{DF}$ tends to 0. We now consider $|\rho_1 - \rho|$ and $|\rho_2 - \rho|$, and claim that

$$|\rho_1 - \rho|, |\rho_2 - \rho| \leq \rho(L_F)^3 \left(1 + 2\rho L_{DF}(L_F)^2\right) - \rho.$$

69 For $|\rho_2 - \rho| = \rho_2 - \rho$, the claim holds thanks to (13). To establish this for $|\rho_1 - \rho|$ requires a small
70 calculation:

$$|\rho_1 - \rho| = \rho - \rho_1 \leq \rho - \frac{\rho}{(L_F)^3} \left(1 - \rho\frac{L_{DF}}{L_F}\right) \qquad \text{(due to (12))}$$

$$\leq \rho(L_F)^3 \left(1 + 2\rho L_{DF}(L_F)^2\right) - \rho \qquad \text{(assuming the claim holds)}$$

$$2\rho \leq \rho(L_F)^3 \left(1 + 2\rho L_{DF}(L_F)^2\right) + \frac{\rho}{(L_F)^3} \left(1 - \rho\frac{L_{DF}}{L_F}\right)$$
$$\text{(reformulating the previous inequality)}$$

$$2 \leq (L_F)^3 + \frac{1}{(L_F)^3} + 2\rho L_{DF}(L_F)^5 - \rho\frac{L_{DF}}{(L_F)^4},$$

71 where the final inequality holds because $x^3 + x^{-3} \geq 2$, and $2x^5 - x^{-4} \geq 0$, for $x \geq 1$. We now
72 consider the function

$$f(\delta) = \rho^2 + \rho^2(1+\delta)^2 - 2\rho^2(1+\delta)\sqrt{1-(\varepsilon_2)^2}$$
$$= \rho^2 \left(\delta^2 + 2\left(1 - \sqrt{1-(\varepsilon_2)^2}\right)\delta + 2\left(1 - \sqrt{1-(\varepsilon_2)^2}\right)\right).$$

73 The function $f$ is a second order polynomial in $\delta$ and because all coefficients are positive, the
74 maximum of $f$ on the interval $[-\delta_m, \delta_m]$ is a attained at $\delta_m$, that is,

$$\sup_{\delta \in [-\delta_m, \delta_m]} f(\delta) = f(\delta_m).$$

75 By combining these results, we see that

$$|c - c'|$$
$$\leq \sqrt{f\left((L_F)^3\left(1 + 2\rho L_{DF}(L_F)^2\right) - 1\right)} + \varepsilon_1$$
$$= \sqrt{\rho^2 + \left(\rho(L_F)^3\left(1 + 2\rho L_{DF}(L_F)^2\right)\right)^2 - 2\rho\left(\rho(L_F)^3\left(1 + 2\rho L_{DF}(L_F)^2\right)\right)\sqrt{1-(\varepsilon_2)^2}}$$
$$+ \varepsilon_1$$
$$= \rho\sqrt{1 + (L_F)^6\left(1 + 2\rho L_{DF}(L_F)^2\right)^2 - 2(L_F)^3\left(1 + 2\rho L_{DF}(L_F)^2\right)\sqrt{1-(\varepsilon_2)^2}} + \varepsilon_1.$$

76 Because both $f(\delta)$ and the bound (13) are monotone in $\rho$, and $\rho$ is bounded by the radius $r$ of the
77 bounding sphere $S(r)$, we conclude that

$$|c - c'|$$
$$\leq 2r\sqrt{1 + (L_F)^6\left(1 + 4rL_{DF}(L_F)^2\right)^2 - 2(L_F)^3\left(1 + 4rL_{DF}(L_F)^2\right)\sqrt{1-(\varepsilon_2)^2}} + \varepsilon_1. \quad (14)$$

78 For every point $c$ in $\mathrm{ax}(\mathcal{S})$ we have found a point $c'$ in $\overline{\mathrm{ax}(F(\mathcal{S}))}$ whose distance is bounded by (14),
79 and therefore the one-sided Hausdorff distance between the two medial axes $\mathrm{ax}(\mathcal{S})$ and $\overline{\mathrm{ax}(F(\mathcal{S}))}$
80 is bounded by the same quantity. Because the symmetrical formulation of the statement, the same
81 bound holds for the Hausdorff distance. $\qquad\square$

82 *Proof of Theorem 4.1* We denote $L_\varphi = \mathrm{Lip}(\phi)$. Expressions (5), (6) and (7) of the main article
83 yield:

$$L_\varphi \leq r\varepsilon, \qquad L_{DF} = \varepsilon, \qquad L_F \leq 1 + L_\varphi \leq 1 + r\varepsilon, \qquad \varepsilon_1 \leq r^2\varepsilon, \qquad \varepsilon_2 \leq r\varepsilon. \quad (15)$$

84 We deduce

$$r\varepsilon \leq 1/4 \implies r\varepsilon(1+r\varepsilon)^2 \leq 1/2 \implies rL_{DF}(L_F)^2 \leq 1/2.$$

Thus, the conditions of Theorem 3.9 are satisfied. Next, we reformulate the inequality (3) of Theorem 3.9.The expression $E$ under the square root at the right hand side of this inequality is:

$$E = 1 + (L_F)^6 \left(1 + 4rL_{DF}(L_F)^2\right)^2 - 2(L_F)^3 \left(1 + 4rL_{DF}(L_F)^2\right) \sqrt{1 - (\varepsilon_2)^2}.$$

By replacing $L_F$ by $1 + L_\varphi$ in $E$, the constants, as well as the degree-one terms in $L_\varphi$, $rL_{DF}$, and $\varepsilon_2$, cancel out. More precisely,

$$E = 16r^2 L_{DF}^2 + r^2 \varepsilon_2^2 + 24rL_\varphi L_{DF} + 9L_\varphi^2 + \mathcal{O}(|(rL_{DF}, L_\varphi, \varepsilon_2)|^3). \tag{16}$$

Finally, by substituting inequalities (15) into (16), we obtain

$$E \le 50r^2\varepsilon^2 + \mathcal{O}\left(r^3\varepsilon^3\right),$$

and

$$d_H(\mathrm{ax}(\mathcal{S}), \mathrm{ax}(F(\mathcal{S}))) \le \left(1 + \sqrt{50}\right) r^2\varepsilon + \mathcal{O}\left(r^3\varepsilon^2\right).$$

$\square$

# B   Federer's tubular neighbourhood lemma

**Lemma B.1 (Federer's tubular neighbourhood lemma, Theorem 4.8 (12) of [19]** *] Let $p \in \mathcal{S}$ and $\mathrm{lfs}(p) > 0$. The generalized normal space to $\mathcal{S}$ at $p$ is characterized by the following property: For any $\rho \in \mathbb{R}$ satisfying $0 < \rho < \mathrm{lfs}(p)$,*

$$\mathrm{Nor}(p, \mathcal{S}) = \{\lambda v \in \mathbb{R}^d \mid \lambda \ge 0, |v| = \rho, \pi_\mathcal{S}(p + v) = \{p\}\}.$$

*In particular, $\mathrm{Nor}(p, \mathcal{S})$ is a convex cone. The generalized tangent space $\mathrm{Tan}(p, \mathcal{S})$ is the convex cone dual to $\mathrm{Nor}(p, \mathcal{S})$.*