# OpenReview forum: "The medial axis of closed bounded sets is Lipschitz stable with respect to the Hausdorff distance under ambient diffeomorphisms"
_NeurIPS.cc/2023/Conference — Submitted to NeurIPS 2023_

### Official Review · Reviewer_WGuX · 2023-07-05

**Soundness:** 3 good
**Presentation:** 3 good
**Contribution:** 2 fair
**Rating:** 3
**Confidence:** 2

**Summary:**

This paper is an extension of a result from Chazal and Soufflet, which states that the Hausdorff distance of a set to its medial axis is lipschitz-bound under ambient deformations. The authors extend the result from C2 sets and C2 deformations to arbitrary closed sets and C1,1 diffeomorphisms which preserve a bounded sphere in the set.

**Strengths:**

Medial axis play a central role in vision, and 3D geometry and investigating may lead to novel approaches and algorithms.

**Weaknesses:**

Unfortunately, I do not believe it is within my capacity to evaluate the full correctness of the theorems presented in this paper, as such I feel uncomfortable to recommend its acceptance. I am a first-time reviewer hence will reassess my thoughts given other reviewers' input, but in a sense, placing the proofs in the supplementary is a somewhat odd choice to me, as the paper is a completely theoretical paper with its crux being the proofs themselves. Additionally, I find the theorem somewhat esoteric, being both a slight extension of Chazal and Soufflet, and requiring the ambient deformation preserve a bounded sphere to only yield a lipschitz bound. Given the above, I find this paper more suiting to a computational geometry/mathematical journal than NeurIPS.

**Questions:**

Can you describe some applied experiment (in, e.g., learning, vision, geometry) you would perform to show the argued practical impact of this result?

**Limitations:**

Yes

---

> ### Author Rebuttal · Authors · 2023-08-04
>
> Q1:
> We prove that certain algorithms involving the medial axis are correct under reasonable assumptions. The result is relevant for any algorithm whose input consists of images acquired using imperfect lenses. Our result allows to quantify the impact of the imperfection. In practice, many machine learning algorithms (in particular for shape recognition) based on the medial axis are already used [10, 18, 25, 33, 41] (as cited in the introduction), for example for the study of root systems of plants. In this context our quantified result can be used to improve the accuracy of such algorithms.

---

> > ### Comment · Area_Chair_Q7eN · 2023-08-17
> >
> > Can the reviewer please take a look at [this comment](https://openreview.net/forum?id=T47mUw8pW4&noteId=bBym4gFu57) to see if your question on application is better addressed?

---

### Official Review · Reviewer_Ynrq · 2023-07-06

**Soundness:** 3 good
**Presentation:** 2 fair
**Contribution:** 2 fair
**Rating:** 4
**Confidence:** 1

**Summary:**

The medial axis of a closed set $\mathcal{S} \subset \mathbb{R}^d$ is defined to be the set of points in $\mathbb{R}^d$ which do not have a unique closest point on $\mathcal{S}.$ The authors develop a notion of stability for such sets with respect to ambient diffeomorphisms of $\mathbb{R}^d.$ The main result proves stability with respect to $C^{1,1}$ diffeomorphisms under additional assumptions about the set $\mathcal{S}$ (for instance, that $\mathcal{S}$ is bounded, See Assumption 3.8 for a full list.) This result is considered a generalization of an earlier result, which makes stronger smoothness assumptions (namely $C^2$) on both the set $\mathcal{S}$ and the class of ambient diffeomorphisms. The authors argue in the early parts of the paper that this extra generality is needed for (unspecified) applications in astrophysics.

**Strengths:**

The expository style in the early parts of the paper is inviting, where it does a good job of illustrating some basic notions, including familiar ones such as the medial axis and less-familiar ones like the generalized tangent space.

**Weaknesses:**

There are a few different criticisms one can make of this paper:

  1. The topic is niche for a NeuRIPS audience.
  2. The main result is technical and difficult for non-experts to verify.
  3. The main result, as described in the introduction, is a marginal improvement over the current state of the art in reference [13], in the sense that one gains only less restrictive assumptions about the regularity of the functions and shape that appear.
  4. The connection to applications is tenuous at best, and no experiments are provided.

With regards to 1 and 2 above, let me draw attention to the statement of the main result in Theorem 3.9 and the preceding assumptions it requires, which take nearly a page to write down even with many prerequisite definitions that appear before it. One would at least hope based on the promises of the introduction that a simple definition of "stability" would be available for use in the statement of the theorem. With regards to 3 and 4, there is very little given to convince the reader that the $C^2$ results are insufficient for applications.

I would not necessarily suggest that a paper with one or more of these deficiencies be excluded from NeuRIPS. However, given that all four issues are present, it seems better to focus on resolving some of them, or to send the paper to another venue (eg. a pure mathematics journal) where some of these criteria are judged to be less important.

**Questions:**

It would be very helpful for readers to point out where to find your main result early on in the introduction. That would be Theorem 3.9, right?

lines 112: I don't understand why the assumption $\mathcal{S}$ and its medial axis being bounded is not a further restriction needed to state your result. There is nothing in Remark 2.1 that addresses the case of an unbounded set $\mathcal{S},$ nor anywhere else in the paper. Moreover, these assumptions do appear in the statement before Theorem 3.9. Thus, it seems incorrect for you to state your theorem in the introduction simply for "closed sets" without further qualification.

---

> ### Author Rebuttal · Authors · 2023-08-04
>
> Q1:
> The main result may differ depending on the audience. For guarantees on a specific algorithm Theorem 3.9 is indeed the most relevant. However, for a general statement about stability and computability of the medial axis, the formulation as given in Theorem 4.1 is more useful.
>
> Q2:
> See the answer to question 1 of the reviewer zm1a:
> "The assumption is technical in one sense, but necessary in another. Let us explain this:
> If we consider just two points in the plane, let’s say $(0,0)$ and $(0,1)$ then the medial axis is a horizontal line. If you perturb $(0,1)$ into $(\sin \theta, \cos \theta)$ the medial axis will have an angle of $\theta$ with the horizontal line (see also Figure 2 in the paper). The Hausdorff distance between two non-parallel lines is infinite, so it is impossible to give a bound on the distance between the medial axes without localizing in one way or another. However, if we restrict ourselves to a ball around the origin of size $r/2$ then the Hausdorff distance between the two restricted medial axes is $\mathcal{O} (r \cdot \theta)$.
> This shows that some bounding is necessary to obtain quantitative results.
>
>
> The assumptions on the ambient diffeomorphism (namely that it keeps the bounding sphere invariant) could be replaced by other assumptions that guarantee localization (as we tried to explain in Remark 2.1):
> For a given point $x$ in $\mathbb{R}^d$ we only have to consider those points of the set $\mathcal{S}$ that are relatively close (a distance at most $r/2$) to $x$ (if they exist). In other words, the medial axis of $\mathcal{S}$ will not be influenced by points that are far away: the bounding sphere of radius $r$ can be ignored (technically one can interpolate between the given diffeomorphism and a diffeomorphism that is the identity beyond bounding sphere). So if the set $\mathcal{S}$ is sufficiently dense (i.e. for every $x$ there there are points in $\mathcal{S}$ that are not further than $r/2$ away from $x$) or locally (for points not so far from the set $\mathcal{S}$) all the stability results go through. In particular, if we consider our first example and we look at a neighbourhood of size $r/2$ of the origin then the stability bounds on the medial axis will hold in this neighbourhood.
>
> We intend to extend our explanation near Remark 2.1 in the final version, because we agree that we were too terse."

---

> > ### Comment · Reviewer_Ynrq · 2023-08-12
> >
> > I am happy with the authors' responses. I will need to read the paper more and monitor discussions before making a final decision with regards to a rating.

---

> > > ### Comment · Area_Chair_Q7eN · 2023-08-17
> > >
> > > Can the reviewer please take a look at [this comment](https://openreview.net/forum?id=T47mUw8pW4&noteId=bBym4gFu57) to see if your question on application is better addressed?

---

### Official Review · Reviewer_JoRT · 2023-07-06

**Soundness:** 3 good
**Presentation:** 3 good
**Contribution:** 3 good
**Rating:** 6
**Confidence:** 4

**Summary:**

The authors prove that the medial axis of closed set is Hausdorff stable without any further assumption on it. In this proof, the authors achieve stability without pruning the medial axis which is a significant advantage. Meanwhile, the results hold for sets in arbitrary dimensions.


**Strengths:**

In originality, this work holds for sets in arbitrary dimentions and removes the limitation of manifold assumption when in proof, and it does not need to prune the medial axis which is a significant advantage.

The quality and clarity is good enough, it is easy to understand the motivation, outline and contribution.

The proof in this paper implies that the medial axis of an imprecise shape is stable. The medial axis plays an important role in the field of computational geometry, computer vision and graphics.

**Weaknesses:**

The result of this work shows the numerical stability of medial axis, but there is little analysis about the impact from noise size and quantity in real world data.


**Questions:**

1. In real world data, the noise quantity and size maybe different, does the result in this work mean that the stability of medial axis of different noisy data is always guaranteed?

2. What does the meaning of `considered set` in line 37 and 112? It seems that the word `considered` is not necessary there.

3. There are some standard examples of the instability of the medial axis mentioned in line 38. Could you give more explanation why these kinds of instability exist? Is the instability essential or numerical?


**Limitations:**

From line 36 to 39, the authors give the limitation about the work, but it could be more clear if there is more explanation. I also asked the questions about it in section Question.

---

> ### Author Rebuttal · Authors · 2023-08-04
>
> Q1:
> If the noise is due to some smooth deformation by e.g. a non-perfect lens, then the answer is yes. However, if you sample from a smooth object, it may be better to prune your axis. Stability results in the latter setting can be found in [29].
>
> Q2:
> We agree with the reviewer. Perhaps it would have been better to just write `the set $\mathcal{S}$'.
>
> Q3:
> The instability of the medial axis is essential, and therefore has a significant numerical impact. The instability makes numerical computation unreliable, unless you are very careful about the perturbations (discussed in this paper) or the pruning (discussed in various other works).
>
> The intuition behind the existence of the instability is as follows: Roughly speaking, the medial axis is sensitive to "curvature"-like effects and global effects. Small (in terms of the Hausdorff distance) local perturbations can still have huge effects on the "curvature" and thus on the shape of the medial axis. The simplest example is the following: We start with two parallel lines. The medial axis is the line  between them right in the middle. Then we perturb one of the lines a tiny bit (in Hausdorff distance) to  create a small bump (with high curvature). As a consequence, the medial axis gains a large new branch that extends towards the bump.
>
> We will add some extra explanation and a figure in the final version, based on this example.

---

> > ### Comment · Reviewer_JoRT · 2023-08-16
> >
> > Thanks for the authors' response, I have no other question, I will make a final decision based on all the discussions and the revised paper.

---

### Official Review · Reviewer_zm1A · 2023-07-07

**Soundness:** 3 good
**Presentation:** 3 good
**Contribution:** 3 good
**Rating:** 6
**Confidence:** 4

**Summary:**

This work proves that the medial axis of closed sets is Hausdorff stable, this extends existing stability result on the stability of the medial axis of C^2 manifolds under C2 ambient diffeomorphisms. The contributions are

1. This work makes no assumptions of the set except the closedness. The stability of the medial axis of smooth manifolds has been intensively studied in the literature, this work omits the manifold assumption.

2. The stability is achieved without pruning the medial axis. Large body of works have to prune the medial axis.

3. The stability results hold for sets in arbitrary dimensions and are insensitive to the dimension of the set itself.

This theoretical result plays a fundamental role in many fields, the generalization is important  to many practical applications.

**Strengths:**

The theoretic results are much general, it doesn't require the manifold assumption, it doesn't need to prune the medial axis, the results hold for any dimensions.

The work is clearly represented. All the key concepts are explained thoroughly, the lemmas, theorems, corollaries are explained in detail, and rigorously formulated. The proofs are step by step, clean and easy to follow.

**Weaknesses:**

The theoretical results are elegant and convincing. It will be helpful to give some numerical experimental results.

**Questions:**

1. The stability result is established with respect to smooth diffemorphisms of the ambient space, is the bounding sphere preserving condition intrinsically essential or technically necessary ? To what extent, one can remove this restriction ?

2. Suppose the set S is a C2 surface, if S is deformed to generate a curvature singularity, the surface becomes C1 at the singularity, the medial axis may change drastically. Please explain why the ambient diffeomorphism C^{1,1} can avoid this situation.

3. For the conjecture, the cut locus is closely related to the sign of the Gaussian curvature on the surface. Small perturbation changing the sign of the curvature may generate conjugate points suddenly. From this point of view, it seems the stability of cut locus may be hard to achieve. Please explain your insights for the conjecture.

**Limitations:**

The current stability result assumes the diffeomorphisms is a small perturbation of the identity, and it preserves the bounding sphere. This constraint seems to be artificial and inconvenient for practical applications. Maybe this requirement can be weakened or the bounding sphere is pushed to infinity.

---

> ### Author Rebuttal · Authors · 2023-08-04
>
> Q1:
> The assumption is technical in one sense, but necessary in another. Let us explain this:
> If we consider just two points in the plane, let’s say $(0,0)$ and $(0,1)$ then the medial axis is a horizontal line. If you perturb $(0,1)$ into $(\sin \theta, \cos \theta)$ the medial axis will have an angle of $\theta$ with the horizontal line (see also Figure 2 in the paper). The Hausdorff distance between two non-parallel lines is infinite, so it is impossible to give a bound on the distance between the medial axes without localizing in one way or another. However, if we restrict ourselves to a ball around the origin of size $r/2$ then the Hausdorff distance between the two restricted medial axes is $\mathcal{O} (r \cdot \theta)$.
> This shows that some bounding is necessary to obtain quantitative results.
>
>
> The assumptions on the ambient diffeomorphism (namely that it keeps the bounding sphere invariant) could be replaced by other assumptions that guarantee localization (as we tried to explain in Remark 2.1):
> For a given point $x$ in $\mathbb{R}^d$ we only have to consider those points of the set $\mathcal{S}$ that are relatively close (a distance at most $r/2$) to $x$ (if they exist). In other words, the medial axis of $\mathcal{S}$ will not be influenced by points that are far away: the bounding sphere of radius $r$ can be ignored (technically one can interpolate between the given diffeomorphism and a diffeomorphism that is the identity beyond bounding sphere). So if the set $\mathcal{S}$ is sufficiently dense (i.e. for every $x$ there there are points in $\mathcal{S}$ that are not further than $r/2$ away from $x$) or locally (for points not so far from the set $\mathcal{S}$) all the stability results go through. In particular, if we consider our first example and we look at a neighbourhood of size $r/2$ of the origin then the stability bounds on the medial axis will hold in this neighbourhood.
>
> We intend to extend our explanation near Remark 2.1 in the final version, because we agree that we were too terse.
>
> Q2:
> The composition of a $C^2$ map with a $C^{1,1}$ map is itself $C^{1,1}$, so the surface can never be just $C^1$. One can associate some curvature to sets of positive reach (this is a non-trivial theory that goes back to Federer, for a complete modern introduction see [RZ19]) and the curvatures of these sets are in a certain sense bounded.  In particular, sets of positive reach cannot have a curvature singularity (and $C^{1,1}$ maps preserve the positivity of reach).
>
> [RZ19] Jan Rataj and Martina Zähle. Curvature measures of singular sets. Springer, 2019.
>
> Q3:
> We are not completely sure that we understand the question of the reviewer.
> If the reviewer means: What happens if the curvature of (for example) a curve changes from slightly positive to slightly negative? In that case, the medial axis is very far away from the curve and thus excluded by our localization assumptions, see the answer to question 1.
> If our reading of the question is not correct, we kindly ask the reviewer to specify in particular what they mean by the cut locus, why the focus lies on the Gaussian curvature, and which conjecture they refer to.

---

> > ### Comment · Area_Chair_Q7eN · 2023-08-20
> >
> > Dear reviewer,
> >
> > The author-reviewer discussion period ends in 2 days. Please review the authors' rebuttal and engage with them if you have additional questions or feedback. Your input during the discussion period is valued and helps improve the paper.
> >
> > Thanks, Area Chair

---

### Official Review · Reviewer_D995 · 2023-07-26

**Soundness:** 3 good
**Presentation:** 3 good
**Contribution:** 3 good
**Rating:** 4
**Confidence:** 1

**Summary:**

This paper proves the Hausdorff stability of the medial axis of closed bounded sets. This is a mathematics paper. The authors set up a foundation of their problem, then applied Theorem 2.6 (from [19]) to complete their proof. The end result is quite beautiful in fact. The authors also show that the results in [13] is a special case of their result.

**Strengths:**

- The paper is written well. Despite not having a mathematics background, I am able to read and understand the majority part of the proof. (Nitpick: there are small typos, for example, some \pi_{S}(p_4) are annotated incorrectly in Figure 1.)
- The authors proved a difficult result (as an indication, [13] is a special case of the result). The proof seems to be correct to me.

**Weaknesses:**

- I have a hard time understanding how this result can be used in machine learning / computer vision / computational geometry applications. Yet the motivation is explained in ln45 - ln73. However, I still do not see how this result can be applied. For the benefit of the readers, I think applications need to be demonstrated, otherwise Neurips might not be the right audience.

**Questions:**

I'd like to understand how this result can be applied in applications that are of interest to Neurips audience.

---

> ### Author Rebuttal · Authors · 2023-08-04
>
> Q1:
> Many machine learning algorithms (in particular for shape recognition) based on the medial axis are already used in practice [10, 18, 25, 33, 41] (as cited in the introduction), for example for the study of root systems of plants. Our paper gives a theoretical underpinning of these results. We show that the features extracted in these papers are stable and therefore reliable and explainable.
> We believe that this paper will be of interest to those at NeurIPS that are interested in explainable A.I. and provably correct algorithms.

---

> > ### Comment · Area_Chair_Q7eN · 2023-08-17
> >
> > Can the reviewer please take a look at [this comment](https://openreview.net/forum?id=T47mUw8pW4&noteId=bBym4gFu57) to see if your question on application is better addressed?

---

### Official Review · Reviewer_wGEh · 2023-07-31

**Soundness:** 3 good
**Presentation:** 3 good
**Contribution:** 3 good
**Rating:** 7
**Confidence:** 3

**Summary:**

In this paper, the authors analyze the stability of the medial axis of a set S, when S is perturbed by a map that is lipschitz with lipschitz derivatives.

This stability result is of interest in numerous applications in machine learning, such as astrophysics.

The author's results improve upon an existing result by Chazal and Soufflet in a few ways:
1. The authors remove an assumption that the set S must be a piecewise smooth manifold; here they only require S to be closed and bounded.
2. They do not require pruning the medial axis.
3. Their result holds in high dimensions.

**Strengths:**

I think the result is significant, and of interest to the neurips community. Compared to Chazal and Soufflet, I think another significant aspect of this result is that this result is quantitative whereas Chazal and Soufflet's result is only qualitative.

**Weaknesses:**

I have some questions about how this paper's results compare to existing results, as well as about several aspects of the result (see below). These may not be considered weaknesses if the authors can address them.

**Questions:**

I have the following questions regarding this result:

1) How significant is removing the manifold assumption? Even if a set is not smooth, can it not be made smooth via some infinitesimally small perturbation (e.g. a gaussian convolution)? Can the authors elaborate on the strength of their result in the context of an application? E.g. for the astrophysics image example, can we not simply smooth the image via a infinitesimal smoothing operation, and then apply Chazal and Soufflet?

2) Is there any existing quantitative bounds that the authors can compare to (even if assumptions differ)? If there are, how does the rate in Theorem 3.9 (line 212) compare to existing rates?

3) To double check, on line 212, as L_F -> 1, L_{DF}-> 0, epsilon_1 and epsilon_2 -> 0, C_L(r, L_F, L_{DF},eps_1,eps_2) -> 0, is that correct?

4) Can the authors give an interpretation of rch(S) defined on line 99? In particular, for a non-smooth S, can rch(S) be 0?

5) if rch(S)=0, the result due to Federer in Theorme 2.6 becomes vacuous. However, the result in Theorem 3.9 does not seem to depend on rch(S) at all, even though it crucially uses Federer's result. Can the authors explain why this is?

---

> ### Author Rebuttal · Authors · 2023-08-04
>
> Q1:
> The strength of weakening the differentiability assumption is best seem when the set is not a manifold. Consider for example a Y shape in the plane. A (Gaussian) convolution cannot make this into a smooth curve. Such Y branches are common in biology (for example the splitting of branches or roots of a plant or in the structure of cells in i.e. a plant). Note that shape recognition questions in biology inspired Blum to introduce the medial axis (although some earlier authors such as Erdos considered the same set).
> Given this remark we think we should have mentioned some applications in biology as well. We focused on the applications in astrophysics because there it would be apparent that high dimensional results are relevant.
>
> In the astrophysical context (part of our motivation), one could think of a planet being ripped into pieces (creating very irregular i.e. non-smooth pieces) while it falls into a black hole and it is at the same time surrounded by the gas (which is usually present in the accretion disk of a black hole). Another scenario would be some colliding objects, like asteroids (which are themselves not very regular), or several shockwaves or jets hitting each other.
>
> None of these examples are manifolds and are thus beyond the reach of the theory of Chazal and Soufflet, who only consider smooth manifolds without boundary.
>
>
> Q2:
> Chazal and Soufflet [13], Theorem 3.2 and 3.3, do not give quantitative results, they only prove convergence, but not that the convergence is Lipschitz (Theorem 4.1 of our paper). Our result gives explicit Lipschitz constants.
>
> Our bounds are significantly better than the recent contribution in Lieutier and Wintraecken [29] (the most recent paper that gives bounds in the setting where one prunes) which only gives 1/2-Hölder bounds on the Hausdorff distance.
>
> Q3:
> This is indeed correct.
>
> Q4:
> Yes, many non-smooth sets have 0 reach: In fact, Federer [19] says that sets of positive reach are piecewise (in a very weak sense) C^{1,1}, meaning that the derivative is Lipschitz continuous. The simplest example of a set with reach 0 is perhaps two line segments meeting at a non-zero angle.
>
>
> Q5:
> Many thanks for this question, because this was rather a big surprise to us as well. One can give some intuition. It suffices to consider the balls centred on the medial axis, and we apply Federer's result to these balls and not to the original set. These balls are well defined even if the set doesn’t have positive reach. Now, roughly speaking, one has the following: If such a ball is large, meaning that you are far away from the set, then Federer’s result will give stability. However, if the ball is small then after applying the map $f$ it will remain a slightly deformed small ball and (the point of) the medial axis (we are interested in) has to lie in this ball. To put it differently, because the ball is small the point has nowhere to go.

---

> > ### Comment · Reviewer_wGEh · 2023-08-18
> >
> > Thank you for the response. These address my concerns and I increased my score to 7.

---

### Comment · Area_Chair_Q7eN · 2023-08-14
**Applications of Medial Axis in AI**

Dear Authors,

A number of reviewers (WGuX, Ynrq, D995) have raised concerns about the **relevance of medial axis to current AI research**. I see that you have listed a number of works [10, 18, 25, 33, 41]  in your response and in your paper. However, it is not immediate clear how relevant these papers are to *current research* in AI. For instance, [25, 33] are quite old. It is certainly possible that these are very relevant even today, but I believe that the authors can make a much stronger case for their paper if they can provide more details, rather than simply provide a list of papers.

For this reason, I hope that the authors can **briefly describe 2-3 concrete applications of medial axis in recent AI papers, which they find to be the most impactful.**

Simply a summary of the application will do, along with a *very brief* explanation of how medial axis is used. There is *no need* to go into details of why the theory in your submitted paper is specifically relevant.

I understand that it is common in NeurIPS papers to simply list a number of papers as justification for motivation, and that is usually sufficient. However, this particular topic is unfamiliar to many of the reviewers, and it will immensely help our future discussions if the authors can take the time to detail a few concrete applications.

-AC

---

> ### Author Response · Authors · 2023-08-16
> **Applications and context**
>
> SHORT ANSWER:
> We list the following applications of the medial axis in AI research:
> -Chambers and collaborators (cited in the paper) identify various biological deficiencies of plants based on the medial axes of their root systems.
> -A recent work that employs the medial axis for computer vision is [TK11]. The authors develop a top-down object detection and segmentation approach that uses a skeleton-based shape model and that works directly on real images.
> -In [HCC12+] and [LB13], the authors exhibit the role that the medial axis plays in biological (as well as computer) vision. In other words, they unveil the importance of the medial axis in how monkeys recognize objects.
>
> [TK11] Trinh, Nhon H.; Kimia, Benjamin B. Skeleton search: Category-specific object recognition and segmentation using a skeletal shape model. International Journal of Computer Vision, 2011, 94: 215-240.
>
> [HCC12+] Hung, Chia-Chun, Eric T. Carlson, and Charles E. Connor. "Medial axis shape coding in macaque inferotemporal cortex." Neuron 74.6 (2012): 1099-1113.
>
> [LB13] Lescroart, Mark D., and Irving Biederman. "Cortical representation of medial axis structure." Cerebral cortex 23.3 (2013): 629-637.

---

> > ### Author Response · Authors · 2023-08-16
> > **Additional information**
> >
> > LONG ANSWER:
> > Below we mention some more applications and provide more context in the case the reviewer(s) find this helpful.
> >
> >
> > Let us start by giving a quick example of the medial axis, which hopefully illustrates why the medial axis is useful:
> > Suppose we are given the outline of a hand in the plane. Then the medial axis will be a tree-like structure with 5 (main)  branches for the fingers. It is simple (e.g. by using the medial axis transfer)  to split the outline of the hand into the fingers (shape segmentation/ feature extraction). Similarly if one sees a skeleton of a shape with 5 main branches, the ends of four of which split into 5 branches the assumption that it concerns a skeleton the outline of a human is reasonable (shape recognition or learning).
> > We believe that shape segmentation and shape learning are intricately linked, [TK11] says the following:
> > ``A general consensus has emerged in the computer vision community that the once considered distinct fundamental problems of segmentation and object recognition are inherently interdependent.''
> > As a side note the main topic of [TK11] is a hierarchical shape segmentation method based on the medial axis.
> >
> >
> > The application of the medial axis and the medial axis transform for 3D shapes to deep neural networks is quite recent [HWQ+19]. The main contributions of that paper are:
> > ``-We present the first deep neural network architecture that can learn the features of MAT, for 3D object recognition.
> > -By utilizing MAT’s edge information, we design Group-MAT and Edge-Net modules to capture local features from its topological structure, which achieves remarkable performance on 3D shape classification task, even for MATs with very few number of spheres only.
> > -We construct an open MAT dataset: ModelNet40-MAT, by repairing the majority of 3D models in ModelNet40.''
> > This paper has already lead to a number of follow up papers:
> > https://scholar.google.com/scholar?cites=8534566892115821177&as_sdt=2005&sciodt=0,5&hl=en
> > For the application of deep learning architectures for shape understanding in the 2D setting we already referred to [18], published in the same year as [HWQ+19].
> > A more recent learning paper that focuses on skeletonization is [YYH+20]. The applications mentioned there are:  shape recognition, 3D reconstruction, segmentation, shape matching, pose estimation, action recognition, animation, extracting skeletons of 3D shapes predicting curve skeletons for 3D shapes using networks. For a more extensive list of applications related to graphics we refer to [35].
> > A slightly older paper on learning the medial axis is [LWS+15].
> >
> >
> > The medial axis is a method to skeletonize a shape which is (now) provably stable. The topic of skeletonization seems to be still important in the AI community:
> > More general skeletons than the medial axis are also still actively studied by the learning community [XZS+19], see also [32]. In [XZS+19]  the authors use neural networks to learn the skeleton of a shape, this skeleton is based on the medial axis (but not exactly equal to the medial axis). In [SLF14] the authors also try to learn a skeleton with application in medial imaging in mind (recognizing blood vessels), given the emphasis on the distance in the learning method of that paper it seems to us that it must be related to the medial axis, but that paper does not discuss the classical literature. The authors of [SZJ+17] similarly say that they learn a skeleton based on the symmetry set (the symmetry set is a generalization of the medial axis, where instead of looking at balls that are empty and touch the set you look at any ball that is tangent to the set in multiple points, but may have a non-empty interior), however the figures seem to be more compatible with the medial axis and not with the symmetry set. Unfortunately [SZJ+17] is quite applied in nature so that it is difficult to reconstruct the exact definitions. In [ZSG+18] the authors use the medial axis as their definition of the skeleton. There are many other recent skeleton reconstruction papers such as [LHQ20].
> >
> > Skeletons are used for 3D action recognition [KBA+17], in this setting one assumes one is given 3D trajectories of  a skeleton and tries to infer the action of the object. The precise nature of the skeleton is not of prime interest in [KBA+17]. As mentioned skeletons are also used to detect blood vessels [SLF14].
> >
> > Other applications include the recognition of scene contours (that is images where are a just a few lines depicting the scene) [ZDW+19] and implicit representations of geometry [RLS+21] .

---

> > > ### Author Response · Authors · 2023-08-16
> > > **references to the additional information**
> > >
> > >
> > > [HWQ+19] Hu, J., Wang, B., Qian, L., Pan, Y., Guo, X., Liu, L., & Wang, W. (2019, August). MAT-Net: Medial Axis Transform Network for 3D Object Recognition. In IJCAI (pp. 774-781).
> > >
> > > [YYH+20] Yang, B., Yao, J., Wang, B., Hu, J., Pan, Y., Pan, T., Wang, W. and Guo, X., 2020. P2MAT-NET: Learning medial axis transform from sparse point clouds. Computer Aided Geometric Design, 80, p.101874.
> > >
> > > [LWS+15] Li, P., Wang, B., Sun, F., Guo, X., Zhang, C. and Wang, W., 2015. Q-mat: Computing medial axis transform by quadratic error minimization. ACM Transactions on Graphics (TOG), 35(1), pp.1-16.
> > >
> > > [TK11] Trinh, Nhon H.; Kimia, Benjamin B. Skeleton search: Category-specific object recognition and segmentation using a skeletal shape model. International Journal of Computer Vision, 2011, 94: 215-240.
> > >
> > > [KBA+17] Qiuhong Ke, Mohammed Bennamoun, Senjian An, Ferdous Sohel, and Farid Boussaid. A new representation of skeleton sequences for 3d action recognition. In Proceedings of the IEEE conference on computer vision and pattern recognition, pages 3288–3297, 2017.
> > >
> > > [XZS+19] Zhan Xu, Yang Zhou, Evangelos Kalogerakis, and Karan Singh. Predicting animation skeletons for 3d articulated models via volumetric nets. In 2019 International Conference on 3D Vision (3DV), pages 298–307. IEEE, 2019
> > >
> > > [SLF14] Sironi, Amos, Vincent Lepetit, and Pascal Fua. "Multiscale centerline detection by learning a scale-space distance transform." Proceedings of the IEEE Conference on Computer Vision and Pattern Recognition. 2014.
> > >
> > > [ZDW+19] Rezanejad, M., Downs, G., Wilder, J., Walther, D. B., Jepson, A., Dickinson, S., & Siddiqi, K. (2019). Scene categorization from contours: Medial axis based salience measures. In Proceedings of the IEEE/CVF conference on computer vision and pattern recognition (pp. 4116-4124).
> > >
> > > [SZJ+17] Shen, W., Zhao, K., Jiang, Y., Wang, Y., Bai, X., and Yuille, A. "Deepskeleton: Learning multi-task scale-associated deep side outputs for object skeleton extraction in natural images." IEEE Transactions on Image Processing 26.11 (2017): 5298-5311.
> > >
> > > [LHQ20] Liu, J. J., Hou, Q., & Cheng, M. M. (2020). Dynamic feature integration for simultaneous detection of salient object, edge, and skeleton. IEEE Transactions on Image Processing, 29, 8652-8667.
> > >
> > > [ZSG+18] Zhao, K., Shen, W., Gao, S., Li, D., & Cheng, M. M. (2018). Hi-fi: Hierarchical feature integration for skeleton detection. arXiv preprint arXiv:1801.01849.
> > >
> > > [RLS+21] Rebain, Daniel, Ke Li, Vincent Sitzmann, Soroosh Yazdani, Kwang Moo Yi, and Andrea Tagliasacchi. "Deep medial fields." arXiv preprint arXiv:2106.03804 (2021).

---

### Decision · Program_Chairs · 2023-09-21

**Decision:**

Reject

**Comment:**

The theory in this paper is quite a significant improvement over the best known bound, both because it is non-asymptotic, and because it removes numerous assumptions. Unfortunately, the consensus among reviewers seems to be that this work is more suited for submission at a mathematical venue, or a conference more focused on vision or graphics.

Based on the reviews, one suggestion for future draft is to include some experiments. For instance, authors can compute the medial axis of a non-smooth shape under perturbations, to show that it is indeed stable.